# The Neutralization of the Eosinophil Peroxidase Antibody Accelerates Eosinophilic Mucin Decomposition

**DOI:** 10.3390/cells12232746

**Published:** 2023-11-30

**Authors:** Yoshiki Kobayashi, Hanh Hong Chu, Dan Van Bui, Yasutaka Yun, Linh Manh Nguyen, Akitoshi Mitani, Kensuke Suzuki, Mikiya Asako, Akira Kanda, Hiroshi Iwai

**Affiliations:** 1Airway Disease Section, Department of Otorhinolaryngology, Kansai Medical University, Osaka 573-1010, Japan; honghanh87.hmu@gmail.com (H.H.C.); drdan.aic@gmail.com (D.V.B.); yunys@hirakata.kmu.ac.jp (Y.Y.); nguyenli@hirakata.kmu.ac.jp (L.M.N.); mitaniak@hirakata.kmu.ac.jp (A.M.); suzukken@hirakata.kmu.ac.jp (K.S.); asako@takii.kmu.ac.jp (M.A.); akanda@hirakata.kmu.ac.jp (A.K.); iwai@hirakata.kmu.ac.jp (H.I.); 2Allergy Center, Kansai Medical University Hospital, Osaka 573-1010, Japan

**Keywords:** bronchial asthma, eosinophil, eosinophilic chronic rhinosinusitis, eosinophilic mucin, eosinophil peroxidase

## Abstract

Eosinophilic airway inflammation, complicated by bronchial asthma and eosinophilic chronic rhinosinusitis (ECRS), is difficult to treat. The disease may become refractory when eosinophilic mucin associated with eosinophil peroxidase (EPX) and autoantibodies fills in the paranasal sinus and small airway. This study investigated the functional role of an anti-EPX antibody in eosinophilic mucin of ECRS in eosinophilic airway inflammation. Eosinophilic mucin was obtained from patients with ECRS. The effects of the anti-EPX antibody on dsDNA release from eosinophils and eosinophilic mucin decomposition were evaluated. Immunofluorescence or enzyme-linked immunosorbent assays were performed to detect the anti-EPX antibody and its supernatant and serum levels in eosinophilic mucin, respectively. The serum levels of the anti-EPX antibody were positively correlated with sinus computed tomography score and fractionated exhaled nitrogen oxide. Patients with refractory ECRS had higher serum levels of the anti-EPX antibody than those without. However, dupilumab treatment decreased the serum levels of the anti-EPX antibody. Immunoglobulins (Igs) in the immunoprecipitate of mucin supernatants enhanced dsDNA release from eosinophils, whereas the neutralization of Igs against EPX stopped dsDNA release. Furthermore, EPX antibody neutralization accelerated mucin decomposition and restored corticosteroid sensitivity. Taken together, the anti-EPX antibody may be involved in the formulation of eosinophilic mucin and be used as a clinical marker and therapeutic target for intractable eosinophilic airway inflammation.

## 1. Introduction

Chronic rhinosinusitis (CRS) is a common airway inflammatory disease with a high prevalence in Europe and the United States (approximately 10%) [1]. The prevalence of CRS with nasal polyps (CRSwNP) increases up to a few percentages; approximately 60% of CRSwNP exhibit eosinophil-dominant infiltration [2]. Eosinophilic chronic rhinosinusitis (ECRS) is a subtype of CRSwNP characterized by eosinophilic infiltration with type 2 inflammation [3]. In Japan, more than 20,000 patients with ECRS have dysosmia, viscous nasal discharge, and nasal obstruction. Different from non-eosinophilic CRSwNP with neutrophil-dominant infiltration, ECRS is refractory to the combined treatment comprising macrolide with endoscopic sinus surgery (ESS) and often oral corticosteroid-dependent [3]. While intranasal corticosteroids is a treatment option [4,5], its effect is incomplete and transient [6] but effective for allergic rhinitis with type 2 inflammation, mainly in the inferior turbinate. ECRS commonly coexists with bronchial asthma [3]. In particular, more than 80% of severe ECRS cases are associated with severe asthma [7,8,9,10], in which local corticosteroid sensitivity is markedly reduced [11], and there is a strong tendency for recurrence after ESS [8]. Dupilumab is a human monoclonal antibody that directly inhibits interleukin (IL)-4 and IL-13 signaling (a main stream of type 2 inflammation) and is indicated in patients with oral corticosteroid-dependent severe asthma [9,12,13,14] and those with refractory ECRS [15].

Eosinophilic mucin, which is formulated in the presence of eosinophilic peroxidase (EPX), is a predictor for disease severity in ECRS and associated with chronic airway obstruction in severe asthma [16,17]. Given that eosinophils or EPX reduce corticosteroid sensitivity in bronchial epithelial cells, possibly because of the inactivation of phosphatase PP2A, which regulates glucocorticoid receptor (GR) nuclear translocation [18,19,20], corticosteroid resistance may be induced when eosinophilic mucin fills the airway. Therefore, the mechanisms of eosinophilic mucin formulation in airways need to be elucidated to reduce eosinophil mucin congestion and restore corticosteroid sensitivity.

Eosinophil cytolysis, which can also be observed in nasal polyps from patients with ECRS, releases extracellular histone-coated DNA traps (eosinophil extracellular traps, EETs) [21]. EETs are thought to be the major components of eosinophilic mucin [21]. In contrast, neutrophils also release extracellular DNA fibers (neutrophil extracellular traps, NETs). These fibers play a role in infectious or autoimmune diseases [22]. Reduced DNase I activity is associated with the persistence of NETs, in which extracellular DNA could be targeted by the host immune system. Moreover, the production of anti-DNA antibodies inhibits DNase I activity [23]. Extracellular enzymes in NETs, such as myeloperoxidase (MPO), are also recognized by the host immune system, resulting in the production of MPO-antineutrophilic cytoplasmic antibody [24].

In refractory CRSwNP, mucin leads to excessive production of DNA traps, either NETs or EETs, with a high level of cytotoxic granular enzymes [25]. Nuclear-targeted autoantibodies, such as anti-dsDNA immunoglobulin G (IgG), are locally increased in the nasal polyps from refractory CRS [26]. A recent report suggested that the anti-EPX antibody exists in the sputum from patients with severe eosinophilic asthma, which could trigger the release of extensive EETs [27]. This finding suggests that anti-EPX-antibody-induced EETs might be associated with the pathogenesis of refractory ECRS. However, whether the anti-EPX antibody is involved in the formulation of eosinophilic mucin in ECRS remains to be elucidated.

Therefore, this study investigated the functional role of the anti-EPX antibody in eosinophilic airway inflammation.

## 2. Materials and Methods

### 2.1. Cell and Mucin Preparation

Human peripheral blood samples were obtained from healthy volunteers with mild eosinophilia (approximately 4–8% of total white blood cells), and eosinophils were isolated with >98% purity using negative selection via a MACS system and an eosinophil isolation kit (Miltenyi Biotec, Bergish Gladbach, Germany), as previously reported [28]. In this study, we used BEAS-2B, a human bronchial epithelial cell line purchased from the European Collection of Authenticated Cell Culture (Salisbury, UK). Cell viability and toxicity were microscopically evaluated using trypan blue staining and a 3-(4,5-dimethylthiazol-2yr)-2-5-diphenyltetrazolium bromide assay, respectively, as needed. Patients underwent ESS under general anesthesia; 13 mucin samples were obtained from the sinuses of patients with refractory ECRS. The local ethics committee of Kansai Medical University approved this study (approval number: KanIRin1313) and written informed consent was provided by all participants.

### 2.2. Viscoelasticity

For each experiment, the mucin samples were cut into equal-sized pieces and then resuspended in culture media. The viscoelasticity of each sample was measured using a cone-plate viscometer (LVDV2PCP, EKO Instruments, Co., Ltd., Tokyo, Japan) connected to a spindle (CPA-52Z, EKO Instruments) as previous reported [19,29]. In this study, we evaluated mucin decomposition by reducing viscoelasticity and visualization.

### 2.3. DNase I Activity

DNase I activity in cell extracts or serum was assessed using the DNase I Activity Assay kit (Bio Vision, Milpitas, CA, USA) as directed.

### 2.4. dsDNA Measurement

dsDNA levels in culture supernatants of eosinophils were assayed using the QuantiFluor^®^ ONE dsDNA System (Promega, Madison, WI, USA). Briefly, mucin samples were dissolved in phosphate-buffered saline (PBS) containing 0.1% dithiothreitol. The supernatants of mucin were immunoprecipitated with protein A/G agarose. A 96-well plate was coated with immunoprecipitated immunoglobulins (Igs), and then purified peripheral blood eosinophils were added, followed by an incubation overnight in RPMI 1640 with 10% fatal bovine serum at 37 °C in humidified atmosphere with 5% CO_2_. To neutralize autoimmune antibodies against eosinophil granule proteins, Igs on the plate were reacted with eosinophil granule proteins 2 h before the addition of eosinophils, as needed.

### 2.5. Immunofluorescence Staining

Following ESS, mucin was fixed in formalin and embedded in paraffin. After deparaffinization, rehydration, and antigen retrieval via heat treatment, the sections were blocked and added with recombinant EPX (Cloud-Clone, Katy, TX, USA), which was biotinylated using the Biotin-XX Protein Labeling Kit (Jena Bioscience, Jena, Germany) and anti-citrullinated histone H3 (LS Bio, Seattle, WA, USA) as a DNA release marker, followed by Alexa Fluor^®^ 488-conjugated streptavidin (Jackson ImmunoResearch, West Grove, PA, USA) and CF™ 647 goat anti-mouse antibody (Biotium, Hayward, CA, USA). Additionally, control antibodies were included in each experiment and Hoechst staining (Invitrogen, Paisley, UK) was performed. The slides were visualized using an FV3000 confocal microscopes (Olympus, Tokyo, Japan).

### 2.6. Measurement of EPX-IgG

Specific IgGs against EPX in serum, nasal discharge, or mucin were measured using an enzyme-linked immunosorbent assay (ELISA) with recombinant human EPX (BioVision, Milpitas, CA, USA) as the capture protein; biotin-labeled anti-IgG (Southern Biotech, Birmingham, AL, USA) and HRP-conjugated streptavidin were used for detection. Serial dilutions of EPX-IgG (Santa Cruz Biotechnology, Dallas, TX, USA), which can detect human EPX were used as the standard. Briefly, plate bottoms were coated with 1μg/mL of the capture protein in PBS for 72 h at 4 °C. After blocking, the samples were adequately diluted (1 in 500 for serum and 1 in 5 for supernatant of nasal discharge or mucin) into the sample buffer (PBS containing 1% BSA), reacted overnight at 4 °C, and then incubated with biotin-labeled anti-IgG (1 in 10,000-diluted by sample buffer) for 2 h at room temperature.

### 2.7. Participants

ECRS and asthma were diagnosed according to the Japanese Epidemiological Survey of Refractory Eosinophilic Chronic Rhinosinusitis [8] and the Global Initiative for Asthma guidelines [30], respectively. Patients with ECRS undergoing conventional treatment without oral corticosteroids were defined as those with stable ECRS, whereas patients with refractory ECRS who required any biologic therapies were defined as those with active ECRS. Fourteen patients with active ECRS received 300 mg of dupilumab every 2 weeks, and its efficacy was evaluated before and 4 months after treatment. The characteristics of participants are shown in Table 1.

### 2.8. Corticosteroid Sensitivity

BEAS-2B cells pre-treated with dexamethasone (10^−7^ M) for 45 min were stimulated with thymic stromal lymphopoietin (TSLP) (10 ng/mL) and Poly(I:C) (10 µg/mL) overnight. CCL4 in cell medium was measured using a sandwich ELISA kit (R&D Systems, Minneapolis, MN, USA) as directed. Dexamethasone’s ability to inhibit TSLP and Poly(I:C)-induced CCL4 release was determined as corticosteroid sensitivity. 

### 2.9. Protein Phosphatase Activity

Immunoprecipitation targeting PP2A was performed using an anti-PP2A antibody (Santa Cruz Biotechnology, Dallas, TX, USA). Phosphatase activity in immunoprecipitation was assayed using the SensoLyte^®^ MFP Protein Phosphatase Assay Kit (AnaSpec, San Jose, CA, USA), as previously reported [20].

### 2.10. Statistics Analysis

The Mann–Whitney U test or paired *t*-test and Spearman’s rank method were used for comparisons of the two groups of data and for calculating correlation coefficients, respectively. ANOVA with a post hoc test adjusted for multiple comparisons is used to analyze other data, as appropriate. *p* < 0.05 was considered statistically significant. Descriptive statistics were expressed as means ± SEM.

## 3. Results

### 3.1. Eosinophils Are Associated with Mucin in ECRS

The paranasal sinus, predominantly the ethmoid sinus, in patients with ECRS are congested with eosinophilic mucin (Figure 1A) and polyps. The number of eosinophils infiltrating the nasal polyps is positively correlated with the sinus computed tomography (CT) score, as defined by the Lund–Mackay scale (Figure 1B). Interestingly, mucin decomposition was inhibited under co-incubation with a higher eosinophil concentration, whereas it was promoted under co-incubation with a lower eosinophil concentration (Figure 1C). Consistent with these findings, a high eosinophil concentration reduced DNase I activity in BEAS-2B, suggesting that excessive eosinophil concentration may be involved in mucin decomposition defects (Figure 1D).

### 3.2. Mucin Igs Enhanced dsDNA Release from Eosinophils

IL-5, one of the most potent activators of eosinophils, induced dsDNA release from eosinophils, whereas Igs in the immunoprecipitate of mucin supernatants further enhanced dsDNA releases from eosinophils (Figure 2A). The neutralization of Igs against EPX cancelled dsDNA release compared with other eosinophil granule proteins (major basic protein, eosinophil-derived neurotoxin, eosinophil cationic protein) (Figure 2B). Furthermore, the existence of autoimmune antibodies against EPX was confirmed in eosinophilic mucin where DNA release was stained by citrullinated histone H3 (Figure 2C).

### 3.3. EPX-IgG Levels Increase in Mucin and Serum from Patients with Refractory ECRS

The original ELISA system, in which IgG captured by recombinant human EPX was bound to biotin-labeled anti-IgG, was optimized in order to quantitatively detect autoimmune antibodies. In this assay system, neutralization using recombinant EPX reduced the IgG levels of samples in a dose-dependent manner (Figure 3A). Although no significant differences in EPX-IgG levels were observed in the supernatants of nasal secretion obtained from healthy volunteers, patients with ECRS, patients with CRS (non-ECRS), and patients with allergic rhinitis, the EPX-IgG levels in the supernatants of mucin obtained from patients with refractory ECRS were markedly elevated compared with those in the supernatants of nasal secretion from other subjects (Figure 3B). EPX-IgG was also detected in the serum, and EPX-IgG levels in patients with active ECRS were significantly higher than those in healthy volunteers, patients with stable ECRS, patients with CRS (non-ECRS), and patients with allergic rhinitis (Figure 3C). Notably, serum EPX-IgG levels were positively correlated with the sinus CT score defined by the Lund–Mackay scale and fractional exhaled nitric oxide (FENO) (Figure 3D,E).

### 3.4. EPX Addition Accelerates Mucin Decomposition

Interestingly, the addition of recombinant EPX promoted mucin decomposition, which was observed visually (left panel in Figure 4A) and confirmed by a reduction in viscoelasticity (right panel in Figure 4A). EPX dose-dependently decreased the viscoelasticity of eosinophilic mucin (Figure 4B). Mucin decomposition was inhibited during co-incubation with BEAS-2B, which was restored by EPX treatment (Figure 4C). In addition, the low concentration (0.01 μg/mL) of EPX increased DNase I activity in BEAS-2B (Figure 4D), suggesting that EPX may enhance eosinophilic mucin decomposition through the neutralization of EPX-IgG and the enhancement of DNase I activity. Although corticosteroid sensitivity decreased, potentially because of impaired phosphatase PP2A activity in airway bronchial epithelial cells co-incubated with eosinophilic mucin, the decomposition of eosinophilic mucin caused by the addition of EPX restored its ability to inhibit pro-inflammatory cytokines and chemokines, such as TSLP, CCL4, CCL26, and CXCL8, concomitant with PP2A activity (Figure 4E,F).

### 3.5. Dupilumab Accelerates Mucin Decomposition

Dupilumab, an IL-4 receptor-α antagonist, promoted mucin decomposition by reducing viscoelasticity (Figure 5A) and restoring DNase I activity in BEAS-2B (Figure 5B). Four months after dupilumab treatment (300 mg every 2 weeks), the sinus CT score defined by the Lund–Mackay scale and serum EPX-IgG levels were significantly reduced (Figure 5C,D). Notably, the corticosteroid sensitivity in airway epithelial cells stimulated with patients’ serum obtained 4 months after dupilumab treatment was increased compared with that before treatment (Figure 5E).

## 4. Discussion

This study revealed that the anti-EPX antibody exists in eosinophilic mucin, which may be associated with DNA release from eosinophils. The neutralization of the anti-EPX antibody by recombinant EPX promoted mucin decomposition. This finding indicates that the anti-EPX antibody may play a role in eosinophilic mucin formulation. In addition, the anti-EPX antibody was detected in the serum of patients with active ECRS and decreased by dupilumab treatment, suggesting that it can be used as a potential clinical marker for intractable ECRS.

Interestingly, low eosinophil concentration (less than blood concentration levels) accelerated mucin decomposition by enhancing the DNase I activity of airway epithelial cells. In a similar fashion, low EPX concentration (normal range of serum levels [31]) enhanced DNase I activity. In contrast, high eosinophil concentration delayed mucin decomposition by reducing DNase I activity. Furthermore, a high EPX concentration with H_2_O_2_ and KSCN generates covalent disulfide mucin crosslinks [17]. Thus, excessive eosinophil concentration (including high concentration of EPX) might be associated with mucin formulation. On the contrary, physiological levels of eosinophils and low EPX concentration upregulate DNase I activity and possibly neutralize anti-EPX antibody, thereby inhibiting mucin formulation. EPX inhibits mucin release in the airway epithelial cells of hamsters [32], which is consistent with our findings.

During eosinophilic airway inflammation, corticosteroid sensitivity is locally decreased due to the dysfunction of phosphatase, which is involved in GR nuclear translocation [11] and induced by EPX (higher concentration: 10 μg/mL). In the present study, eosinophilic mucin, including the anti-EPX antibody, induced the reduction in corticosteroid response, whereas a lower EPX concentration restored its response, concomitant with phosphatase PP2A activity. This finding suggests that the anti-EPX antibody might be involved in corticosteroid resistance and that the neutralization of the anti-EPX antibody could restore this condition. However, the underlying mechanisms of the corticosteroid resistance of the anti-EPX antibody remain unknown. Although the removal of nasal polyps with a massive amount of eosinophilic mucin by ESS can temporarily normalize the condition in a local area, patients with severe ECRS possess a high risk of recurrence of eosinophilic airway inflammation with mucin formulation [8]. To inhibit the recurrence, the anti-EPX antibody should be neutralized or reduced. Dupilumab is used to treat oral corticosteroid-dependent severe asthma [9,12,13,14] because it inhibits IL-4, a key cytokine involved in the induction of corticosteroid resistance [33]. In the present study, dupilumab increased DNase I activity, which was decreased by co-incubation with eosinophilic mucin in airway epithelial cells. This resulted in mucin decomposition and anti-EPX antibody reduction, which may contribute to the restoration of corticosteroid sensitivity.

TSLP and higher concentrations of EPX synergistically induce eosinophilic chemokines, such as CCL4, CCL5, CCL11, and CCL26, in airway epithelial cells [34]. Importantly, the receptors for these chemokines are also expressed on other type 2 inflammatory cells, including Th2 cells or type 2 innate lymphoid cells (ILC2) [35]. Furthermore, CXCL8 production was enhanced when airway epithelial cells co-existed with eosinophils [36]. CXCL8, a neutrophilic chemokine, also has a mild effect on eosinophil chemotaxis [37]. Aside from eosinophils, Th2 cells, and ILC2, neutrophils are also involved in severe asthma [38] and associated with corticosteroid resistance [39]. The reduction in corticosteroid response due to the existence of eosinophilic mucin enhances the production of these chemokines and the accumulation of eosinophils and other inflammatory cells in the airway. Thus, eosinophilic mucin decomposition is important for managing intractable eosinophilic airway inflammations, such as severe asthma and ECRS. In addition to dupilumab, omalizumab could also promote eosinophilic mucin decomposition and restore corticosteroid sensitivity [19].

In the patients with refractory ECRS, EPX-IgG levels in the supernatants of eosinophilic mucin were two digits higher than those in the serum. Importantly, serum levels of EPX-IgG were positively correlated with the sinus CT score and FENO levels. Serum EPX-IgG levels decreased after treatment with dupilumab, concomitant with clinical effects, suggesting that serum EPX-IgG levels may reflect the clinical condition and be a clinical marker under refractory eosinophilic airway inflammation. 

## 5. Conclusions

After our analysis of the anti-EPX antibody in eosinophilic mucin obtained from patients with refractory ECRS, we conclude that the presence of the anti-EPX antibody delays mucin decomposition, lowers DNase I activity, and induces eosinophil extracellular traps and corticosteroid insensitivity. The neutralization of the EPX antibody accelerates eosinophilic mucin decomposition and could be a novel therapeutic target for restoring corticosteroid sensitivity reduced under eosinophilic airway inflammation.

## Figures and Tables

**Figure 1 cells-12-02746-f001:**
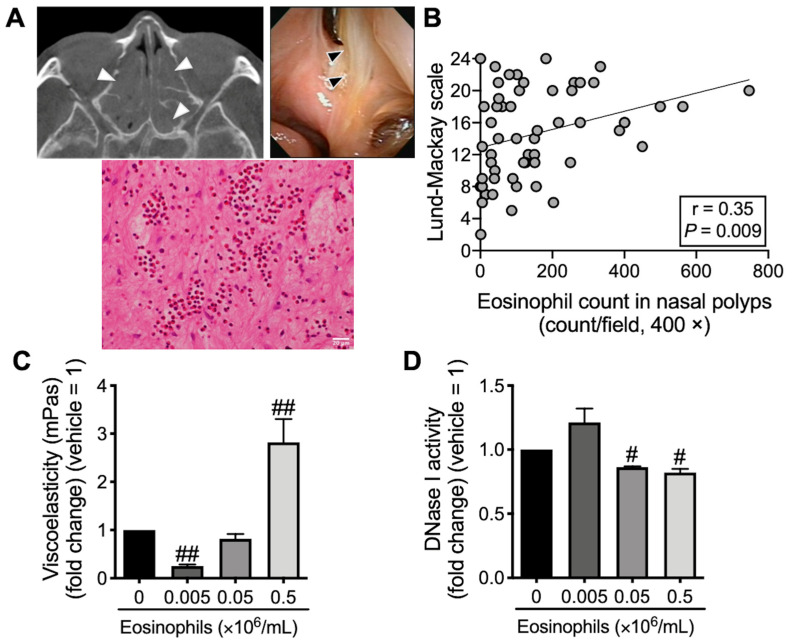
Eosinophilic mucin in ECRS. (**A**) CT scan of the ethmoid sinus (upper-left panel), as well as intranasal endoscopic (upper-right panel) and histological (hematoxylin and eosin staining; 400× magnification, lower panels) findings of mucin. Scale bars in the bottom-right corner indicate 20 μm. Arrow heads indicate mucin. (**B**) Correlation between eosinophil count in nasal polyps and sinus CT score (Lund–Mackay scale) in ECRS. The individual values of 56 subjects are shown. (**C**) Viscoelasticity of mucin was evaluated using a viscometer 72 h after the co-incubation of mucin and purified peripheral blood eosinophils. (**D**) The overnight co-incubation of BEAS-2B cells and purified eosinophils, DNase I in cell extracts of BEAS-2B was assayed. Values represent the means ± SEM values of four experiments. ^#^
*p* < 0.05, ^##^
*p* < 0.01 (vs. without eosinophils).

**Figure 2 cells-12-02746-f002:**
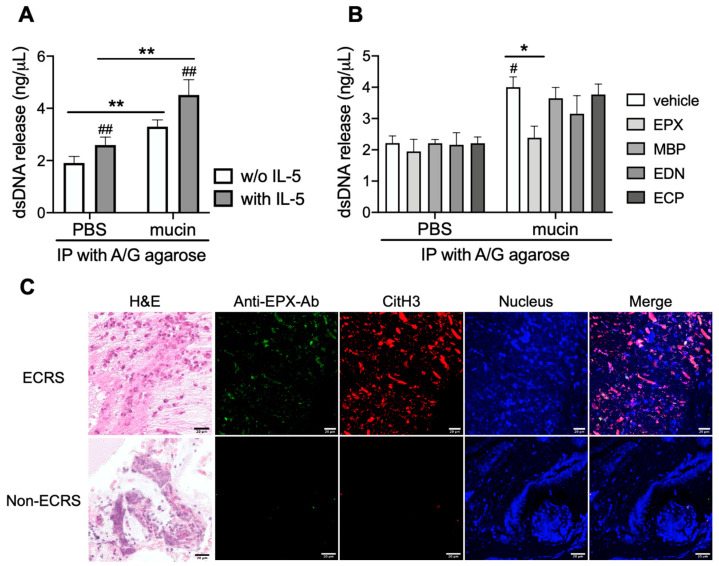
Igs-induced dsDNA release from eosinophils. Purified peripheral blood eosinophils were incubated on the plate coated with Igs in the immunoprecipitate of mucin supernatants. (**A**) Concentrations of dsDNA were released from eosinophils with or without IL-5 (10 ng/mL). (**B**) Igs on the plate were reacted with eosinophil granule proteins (eosinophil peroxidase, EPX; major basic protein, MBP; eosinophil-derived neurotoxin, EDN; eosinophil cationic protein, ECP; 1 µg/mL) 2 h before eosinophils were added. Values represent the means ± SEM values of four experiments. ^#^
*p* < 0.05, ^##^
*p* < 0.01 (vs. PBS without Igs); * *p* < 0.05, ** *p* < 0.01 (as shown between the two groups). (**C**) Autoimmune antibodies against EPX in eosinophilic mucin from patients with ECRS (upper panel). Proteins bound to EPX (anti-EPX antibody, green), citrullinated histone H3 (CitH3, red), and nucleus (blue) are stained with hematoxylin and eosin (H&E). Non-eosinophilic mucin from patients with CRS (non-ECRS) (lower panel) was used as a control. Images are shown at 400-fold magnification, and scale bars in the bottom-right corner of Merge indicate 20 μm. Data are representative of at least three independent experiments.

**Figure 3 cells-12-02746-f003:**
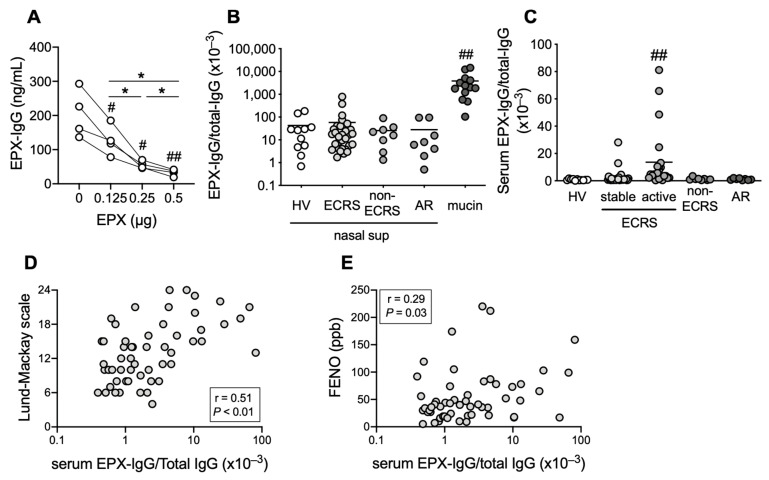
EPX-specific IgG levels in ECRS. (**A**) Neutralization of EPX-IgG in mucin supernatant. EPX-IgG levels measured using our original ELISA system decreased with the addition of recombinant EPX in dose-dependent manner. Individual values were shown (*n* = 4). ^#^ *p* < 0.05, ^##^ *p* < 0.01 (vs. without EPX); * *p* < 0.05 (as shown between the two groups). (**B**,**C**) EPX-IgG levels in supernatant of nasal secretion (nasal sup) or mucin (**B**), and serum (**C**). Individual values in each group represent ratios to total IgG levels. Healthy volunteers (HV, *n* = 8), stable ECRS (eosinophilic chronic rhinosinusitis, *n* = 31), active ECRS (*n* = 23), non-ECRS (*n* = 8), AR (allergic rhinitis, *n* = 8), and mucin (*n* = 13, out of 23 active ECRS). ^##^ *p* < 0.01 (vs. other groups). (**D**,**E**) Correlation between serum EPX-IgG levels and sinus CT score (Lund–Mackay scale) (**D**), and fractional exhaled nitric oxide (FENO) (**E**) in patients with ECRS. Individual values are shown (*n* = 54).

**Figure 4 cells-12-02746-f004:**
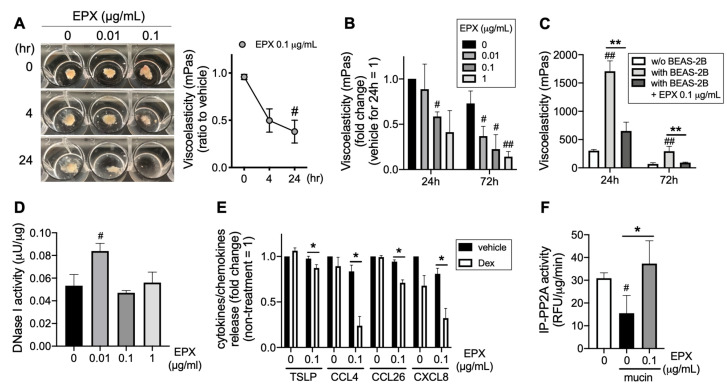
Effects of EPX on mucin decomposition. (**A**–**C**) Effect of EPX on mucin decomposition from patients with ECRS. Eosinophilic mucin was treated with EPX for up to 72 h (**A**,**B**). Mucin co-incubated with BEAS-2B cells was treated with EPX for up to 72 h (**C**). Mucin viscoelasticity was measured using a viscometer. (**D**) Effect of EPX on DNase I activity in airway epithelial cells. BEAS-2B cells were treated with EPX for 72 h. The DNase I activity in BEAS-2B cell extracts was assayed. (**E**,**F**) Effect of EPX on the ability of corticosteroid to inhibit TSLP, CCL4, CCL26, and CXCL8 (**E**), as well as phosphatase activity in immunoprecipitates with PP2A (IP-PP2A). BEAS-2B cells were co-incubated with eosinophilic mucin and EPX for 72 h. Values represent the means ± SEM values of four (**A**,**D**–**F**) or three (**B**,**C**) experiments. ^#^
*p* < 0.05, ^##^
*p* < 0.01 (vs. time = 0 in A, without EPX in **B** and **D**, without BEAS-2B in **C**); * *p* < 0.05, ** *p* < 0.01 (as shown between the two groups).

**Figure 5 cells-12-02746-f005:**
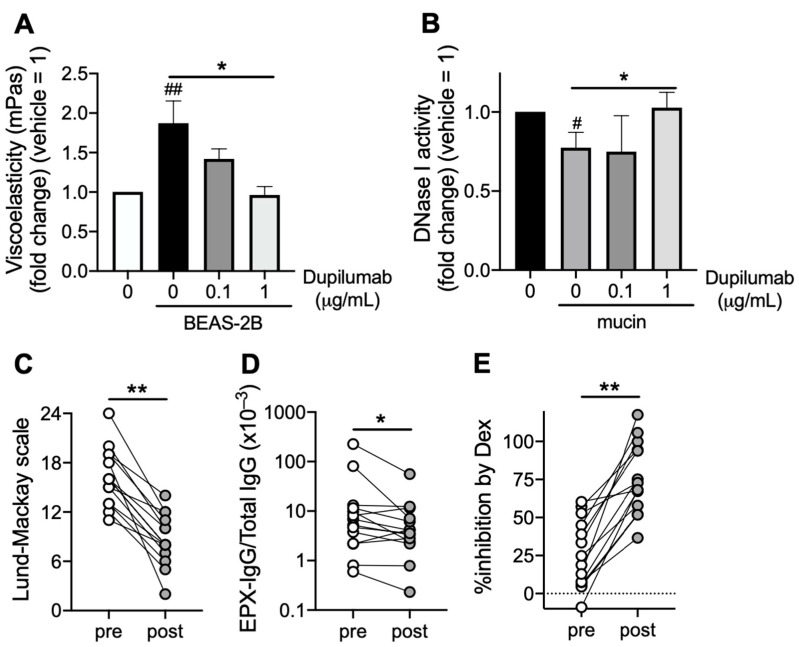
Effect of dupilumab on mucin decomposition. (**A**) Effect of dupilumab on mucin decomposition from patients with ECRS. Eosinophilic mucin co-incubated with BEAS-2B cells was treated with dupilumab for 24 h. The mucin viscoelasticity was measured using a viscometer. Control is mucin alone without BEAS-2B. (**B**) Effect of dupilumab on DNase I activity in airway epithelial cells. BEAS-2B cells co-incubated with eosinophilic mucin were treated with dupilumab for 24 h. DNase I activity in BEAS-2B cell extracts was assayed. Control is BEAS-2B alone without mucin. (**C**,**D**) Sinus CT score (Lund–Mackay scale, **C**) and serum EPX-IgG levels (**D**) were evaluated before (pre) and 4 months after (post) treatment with dupilumab. (**E**) The inhibitory activity by dexamethasone on TSLP and Poly(I:C)-induced CCL4 production (%inhibition by Dex) in BEAS-2B co-incubated with one in 10-diluted patients’ serum overnight was measured. Sera were obtained before (pre) and 4 months after (post) treatment with dupilumab. Values represent the means ± SEM values of four (**A**,**B**) experiments, and individual values (*n* = 14) are shown (**C**–**E**). ^#^
*p* < 0.05, ^##^
*p* < 0.01 (vs. non-treatment control); * *p* < 0.05, ** *p* < 0.01 (as shown between the two groups).

**Table 1 cells-12-02746-t001:** Patient characteristics.

	HV (*n* = 8)	Stable ECRS (*n* = 31)	Active ECRS (*n* = 23)	Non-ECRS (*n* = 8)	AR (*n* = 8)	*p* Value
Age	52.3 ± 8.2	59.5 ± 10.5	54.4 ± 14.8	56.4 ± 15.2	37.1 ± 19.3	*p* = 0.0023
Gender (M/F)	3/5	13/18	12/11	7/1	6/2	*p* = 0.0136
Smoking status (never/ex)	7/1	18/13	17/6	4/4	5/3	*p* = 0.5257
Eosinophils (per μL)	225 ± 109	401 ± 396	753 ± 1069	245 ± 178	196 ± 162	*p* = 0.1011
FEV_1_ (%predicted)	92.4 ± 8.9	74.9 ± 18.7	79.7 ± 22.6	92.8 ± 13.7	90.1 ± 15.2	*p* = 0.0685
FEV1/FVC	83.1 ± 4.7	69.3 ± 12.0	69.6 ± 16.1	74.6 ± 8.2	83.6 ± 4.9	*p* = 0.0075
FEF_25–75_ (%predicted)	83.0 ± 16.5	45.1 ± 19.7	51.0 ± 35.1	61.4 ± 25.4	75.2 ± 20.0	*p* = 0.0016
Treatment						
ICS (μg) ^(1)^	none	800 ± 345	1087 ± 285	none	none	*p* = 0.0005
LABA	0	22	23	0	0	*p* = 0.0716
LTRA	0	18	14	0	4	*p* = 0.8999
Anti-histamine	0	6	12	1	3	*p* = 0.0898
INS	0	19	13	0	3	*p* = 0.5829
OCS	0	0	3 [2.8 ± 0.7 mg] ^(2)^	0	0	*p* = 0.6391
Macrolide	0	3	1	3	0	*p* = 0.4616

AR: allergic rhinitis; ECRS: eosinophilic chronic rhinosinusitis; FEF_25–75_: forced expiratory flow between 25% and 75% of vital capacity; FEV_1_: forced expiratory volume in 1 s; FVC: forced vital capacity; HV: healthy volunteers; ICS: inhaled corticosteroid; INS: inhaled nasal corticosteroid; LABA: long-acting β_2_-adrenergic agonist; LTRA: leukotriene receptor antagonist; OCS: oral corticosteroid, ^(1)^ fluticasone propionate equivalent dose, ^(2)^ prednisolone equivalent dose. *p* values except for ICS indicate differences among groups. *p* value for ICS indicates a difference between stable ECRS and active ECRS.

## Data Availability

Data is contained within the article. Further inquiries can be directed to the corresponding author.

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
