# Peer review of "The Neutralization of the Eosinophil Peroxidase Antibody Accelerates Eosinophilic Mucin Decomposition"

_cells, 2023, doi:10.3390/cells12232746_

Round 1
Reviewer 1 Report
Comments and Suggestions for Authors
General comment:
The authors investigate what role anti-EPX antibody may play in the production of eosinophilic mucin in ECRS and ultimately in eosinophilic inflammation. The authors suggest that this antibody may indeed be involved in eosinophilic mucin and subsequent inflammation, making it a possible biomarker of severe disease.
Specific comments/questions:
Introduction:
· The introduction in general needs more detail, especially with regards to how important it is to solve the problem of refractory eosinophilic inflammation.
o How many people are affected?
o what damage is caused as a result?
o What treatments are normally used, and why aren’t they good enough?
· More detail on how ECRS is different to CRS and allergic rhinitis.
· Authors talk about NETs but not how this is related or relevant to EETs and their work.
Methods:
This section was poorly written and doesn’t give the reader any real sense of what the authors did and why they ran the experiments they did. It’s not enough to write a brief description of the methods as part of the figure legends, a great more detail is needed in this section.
· Cell prep: how many volunteers did you use? Why would a healthy person have mild eosinophilia? Why not isolate and use eosinophils from your target patient group i.e ECRS patients? Can you show that eos from healthy people function/behave the same way as from ECRS patients?
· Viscoelasticity: How did you obtain the mucin samples? Again, how many patients? More details are needed on how this measurement is done, and ref other papers that use the same method.
· dsDNA measurement: not enough detail. It sounds like you are doing 2 different methods in this section, one on the culture supts (what were the culture conditions?) and another method on the mucin supts (how did you get these samples?)
· IF staining: Who was having endoscopic surgery, and why?
· Steroid sensitivity: Its important to measure this not only in the BEAS-2B cell line but also in patient-derived eosinophils.
· Mucin decomposition: There is no description of how this was qualitatively/quantitatively measured.
Results:
As I was unsure of what many of the methods were, it was extremely difficult to follow and evaluate the results to interpret what they might mean.
· 3.1: Authors need arrows in part A of the figure to show what they are describing.
· 3.2: Why have the authors chosen to stimulate with IL-5?
· 3.4: In Figure 4 - For simplicity, maybe group all the expts with BEAS-2B cells together, and all expts with mucin together.
· 3.5: This is the first mention of Dupilumab – this should be briefly discussed in the introduction, and protocols of its use described in the methods.
· Figure 5: what is the difference between the 2 bars labelled as “0” in part A and B of the figure?
Discussion:
· This section needs to be more detailed.
o Discus the results more in depth, how do the results impact disease pathogenesis, how do these findings change current treatment strategies, what other diseases may this mechanism be important in? etc…
Comments on the Quality of English Language
The quality of English is generally good, with a handful of grammatical mistakes.
Author Response
Reviewer #1:
We would like to thank for Reviewer’s favorable comments. We tried to answer the questions.
Comments:
- The introduction in general needs more detail, especially with regards to how important it is to solve the problem of refractory eosinophilic inflammation.
o How many people are affected?
o what damage is caused as a result?
o What treatments are normally used, and why aren’t they good enough?
(Response)
According to reviewer’s suggestions, we added more detail as follows;
Chronic rhinosinusitis (CRS) is a common airway inflammatory disease with a high prevalence in Europe and the United States (approximately 10%). The prevalence of CRS with nasal polyps (CRSwNP) increases up to a few percentage; approximately 60% of CRSwNP exhibit eosinophil-dominant infiltration Eosinophilic chronic rhinosinusitis (ECRS) is a subtype of CRSwNP characterized by eosinophilic infiltration with type 2 inflammation. In Japan, more than 20,000 patients with ECRS have dysosmia, viscous nasal discharge, and nasal obstruction. Different from non-eosinophilic CRSwNP with neutrophil-dominant infiltration, ECRS is refractory to the combined treatment comprising macrolide with endoscopic sinus surgery (ESS) and often oral corticosteroid-dependent. While intranasal corticosteroids is a treatment option, its effect is incomplete and transient but effective for allergic rhinitis with type 2 inflammation mainly in inferior turbinate. ECRS commonly coexists with bronchial asthma. In particular, more than 80% of severe ECRS cases are associated with severe asthma, in which local corticosteroid sensitivity is markedly reduced, and there is a strong tendency for recurrence after ESS.
- More detail on how ECRS is different to CRS and allergic rhinitis.
(Response)
We also mentioned this information in Introduction section (please see Response 1).
- Authors talk about NETs but not how this is related or relevant to EETs and their work.
(Response)
Althogh there are no direct relationship between NETs and EETs, NETs or EETs are excessively producted in mucin of refractory CRSwNP. Further, mucin contains a lot of granular proteins which can be targetted by the host immune system, resulting in production of autoantibodies against these proteins. We mentioned about it in Introduction section.
- Cell prep: how many volunteers did you use? Why would a healthy person have mild eosinophilia? Why not isolate and use eosinophils from your target patient group i.e ECRS patients? Can you show that eos from healthy people function/behave the same way as from ECRS patients?
(Response)
We used four healthy volunteers for each experiment. Because several million eosinophils are needed for each experiment and eosinophils from patients are activated to some extent, donner should be healthy person with mild eosinophilia to uniform the character of eosinophils. Eosinophils from healthy volunteers can be activated by IL-5 stimulation or co-incubation with bronchial epithelial cells. As we mentioned in Materials and Methods section, eosinophils were isolated from the peripheral blood of healthy volunteers with mild eosinophilia (approximately 4%–8% of total white blood cells) to uniform the character of eosinophils.
- Viscoelasticity: How did you obtain the mucin samples? Again, how many patients? More details are needed on how this measurement is done, and ref other papers that use the same method.
(Response)
We refered previous paper regarding the measurement principle. Thirteen mucin samples were obtained when the patients with refractory ECRS recieved endoscopic sinus surgery under general anesthesia. We added this information in Materials and Methods section 2.1. We used the samples from 3 or 4 patients for each experiment. Regarding to the number of samples used for each experiment was indicated in Figure legends, respectively.
- dsDNA measurement: not enough detail. It sounds like you are doing 2 different methods in this section, one on the culture supts (what were the culture conditions?) and another method on the mucin supts (how did you get these samples?).
(Response)
Both are the same method. We measured dsDNA on the eosinophil culthre supernatants. In another experiment, to neutralize auto-immune antibodies against eosinophil granule proteins, immunoglobulins on the plate were reacted with eosinophil granule proteins 2 h before eosinophils were added. We mentioned details in Methods section.
- IF staining: Who was having endoscopic surgery, and why?
(Response)
Mucin samples were obtained when the patients with refractory ECRS recieved endoscopic sinus surgery under general anesthesia. We added this information in Materials and Methods section 2.1.
- Steroid sensitivity: Its important to measure this not only in the BEAS-2B cell line but also in patient-derived eosinophils
(Response)
As reviewer pointed out, it’s important to evalute steroid sensitivity in patient-derived eosinophils. When eosinophils are co-incubated with BEA2-2B cells, their viablility is significantly prolonged (Kobayashi Y, et al., Front Immunol. 2018 Sep 25;9:2192), indicating that eosinophils located in local inflammatory sites are activated and may have resistance to steroid-induced apoptosis. Therefore, we should ideally use eosinophils located in airway inflammatrory sites when we evaluate those steroid sensitity. In this study we focus on steroid sensitivity in airway epithelial cells affected by anti-EPX antibody. We need to plan to evaluate steroid sensitivity in eosinophils located in airway inflammatrory sites as a future work.
- Mucin decomposition: There is no description of how this was qualitatively/quantitatively measured.
(Response)
In this study, we evaluated mucin decomposition by reduction of viscoelasticity and visualization. We mentioned about it in Material and Methods section.
- 3.1: Authors need arrows in part A of the figure to show what they are describing.
(Response)
As reviewer suggested, we added arrow heads in the figures and comments in Figure legends.
- 3.2: Why have the authors chosen to stimulate with IL-5?
(Response)
In this study, we chose IL-5 to activate eosinophils and prolong their cell viability. We added “one of the most potent activators of eosinophils” to first sentense in 3.2 section.
- 3.4: In Figure 4 - For simplicity, maybe group all the expts with BEAS-2B cells together, and all expts with mucin together.
(Response)
To confirm direct effect of EPX to mucin decomposition without any interactions with BEAS-2B, mucin alone was treated with EPX in Figure 4 A and B.
- 3.5: This is the first mention of Dupilumab – this should be briefly discussed in the introduction, and protocols of its use described in the methods.
(Response)
As reviewer suggested, we mentioned about dipulumab in the introduction and described protocol in the methods (2.7. Subjects).
- Figure 5: what is the difference between the 2 bars labelled as “0” in part A and B of the figure?
(Response)
Control in part A is mucin alone without BEAS-2B and control in part B is BEAS-2B alone without mucin, respectively. We added these information in Figure legend.
- This section needs to be more detailed.
o Discus the results more in depth, how do the results impact disease pathogenesis, how do these findings change current treatment strategies, what other diseases may this mechanism be important in? etc…
(Response)
According to reviewer’s suggestion, we edited Disussion section.
Reviewer 2 Report
Comments and Suggestions for Authors
Major comments:
1) Even though you indicate (I believe) that the mechanism is unknown, please speculate on one or more possible mechanisms for the perhaps surprising dose responses, with, for example, eosinophils, EPX, and dupilumab, respectively, in Figure 1c, 4d, and 5a-b.
2) In your experiments EPX promote mucin decomposition but in the model by Dunican et al. (reference No. 8), they have EPX as promoting mucin formation to this reviewer’s understanding. Please discuss this and if you agree that there is a discrepancy, why this may be.
3) Please have a Materials and Methods subsection on Subjects and the inclusion or recruitment criteria for each group. For example, how were “active ECRS” and “stable ECRS” in Supplemental Table 1 defined?
Minor comments:
4) Supplemental Table 1: Please provide p values for differences among or between groups.
5) Introduction, lines 41-42: Please expand the description somewhat of how references No. 9 and 10 show that eosinophils or EPX reduce CS sensitivity in bronchial epithelial cells.
6) Methods, subsection 2.6. Please give more details on the ELISA, including coating concentrations, detection antibody concentrations, buffers, incubation times and temperatures, so the interested reader would be able to repeat this.
7) Please don’t provide more significant figures than what the underlying data can motivate. For example, in Fig. 1b, it should be sufficient to have r = 0.35 and p = 0.009, and similarly elsewhere.
8) Fig. 2 legend: Shouldn’t it be scale bars in the “bottom-right” corner(s), not left?
9) Line 203: What do you mean by “steroid ability”?
10) The manuscript needs linguistic revision. For instance, on line 14 (2nd line of abstract, “is difficult to treatment” should be “is difficult to treat”.
Comments on the Quality of English Language
The manuscript needs linguistic revision. For instance, on line 14 (2nd line of abstract, “is difficult to treatment” should be “is difficult to treat”.
Author Response
Reviewer #2:
We would like to thank for Reviewer’s favorable comments. We tried to answer the questions.
Major comments:
- Even though you indicate (I believe) that the mechanism is unknown, please speculate on one or more possible mechanisms for the perhaps surprising dose responses, with, for example, eosinophils, EPX, and dupilumab, respectively, in Figure 1c, 4d, and 5a-b.
(Response)
As we mentioned in a Discussion section, low eosinophil concentration (less than blood concentration levels) accelerated mucin decomposition by enhancing the DNase I activity of airway epithelial cells. In a similar fashion, low EPX concentration (normal range of serum levels) enhanced DNase I activity. In contrast, high eosinophil concentration delayed mucin decomposition by reducing DNase I activity. Furthermore, a high EPX concentration with H2O2 and KSCN generates covalent disulfide mucin crosslinks. Thus, excessive eosinophil concentra-tion (including high concentration of EPX) might be associated with mucin formulation. On the contrary, physiological levels of eosinophils and low EPX concentration upregulate DNase I activity and possibly neutralize anti-EPX antibody, thereby inhibiting mucin formulation.
Regarding dupilumab, it restored DNase I activity, resulting in accelation of mucin decomposition in dose-dependent manner.
- In your experiments EPX promote mucin decomposition but in the model by Dunican et al. (reference No. 8), they have EPX as promoting mucin formation to this reviewer’s understanding. Please discuss this and if you agree that there is a discrepancy, why this may be.
(Response)
In the model by Dunican et al. high concentration of EPX was used with H2O2 and KSCN to generate covalent disulfide mucin crosslinks whereas EPX (yet at lower concentrations) alone was used to neutralize auto-EPX antibody in our experiments. We added this point in a Discussion section.
- Please have a Materials and Methods subsection on Subjects and the inclusion or recruitment criteria for each group. For example, how were “active ECRS” and “stable ECRS” in Supplemental Table 1 defined?
(Response)
We added “Participants” in a Materials amnd Methods section.
Minor comments:
- Supplemental Table 1: Please provide p values for differences among or between groups.
(Response)
We added p values for differences among groups.
- Introduction, lines 41-42: Please expand the description somewhat of how references No. 9 and 10 show that eosinophils or EPX reduce CS sensitivity in bronchial epithelial cells.
(Response)
Eosinophils or EPX reduce corticosteroid sensitivity in bronchial epithelial cells possibly because of inactivation of phosphatase PP2A which regulates glucocorticoid recepotr nuclear translocation. We added this information.
- Methods, subsection 2.6. Please give more details on the ELISA, including coating concentrations, detection antibody concentrations, buffers, incubation times and temperatures, so the interested reader would be able to repeat this.
(Response)
We added details as follows;
Briefly, plate bottoms were coated with 1mg/mL capture protein in PBS for 72 h at 4 ℃. After blocking, the samples were adequately diluted (1 in 500 for serum and 1 in 5 for supernatant of nasal discharge or mucin) into the sample buffer (PBS containing 1% BSA), reacted overnight at 4 ℃, and then incubated with biotin-labeled anti-IgG (1 in 10,000-diluted by sample buffer) for 2 h at room temperature.
- Please don’t provide more significant figures than what the underlying data can motivate. For example, in Fig. 1b, it should be sufficient to have r = 0.35 and p = 0.009, and similarly elsewhere.
(Response)
As reviewer suggested, we changed these descriptions.
- Fig. 2 legend: Shouldn’t it be scale bars in the “bottom-right” corner(s), not left?
(Response)
As reviewer suggested, we corrected it.
- Line 203: What do you mean by “steroid ability”?
(Response)
We mean “corticosteroid sensitivity” by “steroid ability”.
We replaced to “corticosteroid sensitivity”.
- The manuscript needs linguistic revision. For instance, on line 14 (2nd line of abstract, “is difficult to treatment” should be “is difficult to treat”.
(Response)
Again, our manuscript has been carefully reviewed by an experienced editor whose first language is English and who specialises in editing papers written by scientists whose native language is not English.
Round 2
Reviewer 2 Report
Comments and Suggestions for Authors
The previous comments have been addressed.
Comments on the Quality of English LanguageSome minor spelling mistakes in the new text, e.g., "efficasy", should be "efficacy". Please correct and check throughout.
Author Response
We would like to thank for Reviewer’s favorable comments. I corrected the mistake indicated and checked throughout again.